# Recovering Communities in the General Stochastic Block Model Without Knowing the Parameters

**Emmanuel Abbe**
Department of Electrical Engineering and PACM
Princeton University
Princeton, NJ 08540
eabbe@princeton.edu

**Colin Sandon**
Department of Mathematics
Princeton University
Princeton, NJ 08540
sandon@princeton.edu

## Abstract

The stochastic block model (SBM) has recently gathered significant attention due to new threshold phenomena. However, most developments rely on the knowledge of the model parameters, or at least on the number of communities. This paper introduces efficient algorithms that do not require such knowledge and yet achieve the optimal information-theoretic tradeoffs identified in Abbe-Sandon '15. In the constant degree regime, an algorithm is developed that requires only a lower-bound on the relative sizes of the communities and achieves the optimal accuracy scaling for large degrees. This lower-bound requirement is removed for the regime of diverging degrees. For the logarithmic degree regime, this is further enhanced into a fully agnostic algorithm that simultaneously learns the model parameters, achieves the optimal CH-limit for exact recovery, and runs in quasi-linear time. These provide the first algorithms affording efficiency, universality and information-theoretic optimality for strong and weak consistency in the SBM.

## 1 Introduction

This paper studies the problem of recovering communities in the general stochastic block model with linear size communities, for constant and logarithmic degree regimes. In contrast to [1], this paper does not require knowledge of the parameters. It shows how to learn these from the graph toplogy. We next provide some motivations on the problem and further background on the model.

Detecting communities (or clusters) in graphs is a fundamental problem in networks, computer science and machine learning. This applies to a large variety of complex networks (e.g., social and biological networks) as well as to data sets engineered as networks via similarly graphs, where one often attempts to get a first impression on the data by trying to identify groups with similar behavior. In particular, finding communities allows one to find like-minded people in social networks, to improve recommendation systems, to segment or classify images, to detect protein complexes, to find genetically related sub-populations, or discover new tumor subclasses. See [1] for references.

While a large variety of community detection algorithms have been deployed in the past decades, the understanding of the fundamental limits of community detection has only appeared more recently, in particular for the SBM [1–7]. The SBM is a canonical model for community detection. We use here the notation $\text{SBM}(n, p, W)$ to refer to a random graph ensemble on the vertex-set $V = [n]$, where each vertex $v \in V$ is assigned independently a hidden (or planted) label $\sigma_v$ in $[k]$ under a probability distribution $p = (p_1, \ldots, p_k)$ on $[k]$, and each unordered pair of nodes $(u, v) \in V \times V$ is connected independently with probability $W_{\sigma_u, \sigma_v}$, where $W$ is a symmetric $k \times k$ matrix with entries in $[0, 1]$. Note that $G \sim \text{SBM}(n, p, W)$ denotes a random graph drawn under this model, without the hidden (or planted) clusters (i.e., the labels $\sigma_v$) revealed. The goal is to recover these labels by observing only the graph.

Recently the SBM came back at the center of the attention at both the practical level, due to extensions allowing overlapping communities that have proved to fit well real data sets in massive networks [8], and at the theoretical level due to new phase transition phenomena [2–6]. The latter works focus exclusively on the SBM with two symmetric communities, i.e., each community is of the same size and the connectivity in each community is identical. Denoting by $p$ the intra- and $q$ the extra-cluster probabilities, most of the results are concerned with two figure of merits: **(i) recovery** (also called exact recovery or strong consistency), which investigates the regimes of $p$ and $q$ for which there exists an algorithm that recovers with high probability the two communities completely [7,9–19], **(ii) detection**, which investigates the regimes for which there exists an algorithm that recovers with high probability a positively correlated partition [2–4].

The sharp threshold for exact recovery was obtained in [5,6], showing[1] that for $p = a\log(n)/n$, $q = b\log(n)/n$, $a, b > 0$, exact recovery is solvable if and only if $|\sqrt{a} - \sqrt{b}| \geq \sqrt{2}$, with efficient algorithms achieving the threshold. In addition, [5] introduces an SDP, proved to achieve the threshold in [20, 21], while [22] shows that a spectral algorithm also achieves the threshold. The sharp threshold for detection was obtained in [3,4], showing that detection is solvable (and so efficiently) if and only if $(a - b)^2 > 2(a + b)$, when $p = a/n$, $q = b/n$, settling a conjecture from [2].

Besides the detection and the recovery properties, one may ask about the partial recovery of the communities, studied in [1, 19, 23–25]. Of particular interest to this paper is the case of **strong recovery** (also called weak consistency), where only a vanishing fraction of the nodes is allowed to be misclassified. For two-symmetric communities, [6] shows that strong recovery is possible if and only if $n(p - q)^2/(p + q)$ diverges, extended in [1] for general SBMs.

In the next section, we discuss the results for the general SBM of interest in this paper and the problem of learning the model parameters. We conclude this section by providing motivations on the problem of achieving the threshold with an efficient and universal algorithm.

Threshold phenomena have long been studied in fields such as information theory (e.g., Shannon's capacity) and constrained satisfaction problems (e.g., the SAT threshold). In particular, the quest of achieving the threshold has generated major algorithmic developments in these fields (e.g., LDPC codes, polar codes, survey propagation to name a few). Likewise, identifying thresholds in community detection models is key to benchmark and guide the development of clustering algorithms. However, it is particularly crucial to develop benchmarks that do not depend sensitively on the knowledge of the model parameters. A natural question is hence whether one can solve the various recovery problems in the SBM *without* having access to the parameters. This paper answers this question in the affirmative for the exact and strong recovery of the communities.

## 1.1 Prior results on the general SBM with known parameters

Most of the previous works are concerned with the SBM having symmetric communities (mainly 2 or sometimes $k$), with the exception of [19] which provides the first general achievability results for the SBM.[2] Recently, [1] studied fundamental limits for the general model $SBM(n, p, W)$, with $p$ independent of $n$. The results are summarized below. Recall first the recovery requirements:

**Definition 1.** *(Recovery requirements.)* *An algorithm recovers or detects communities in $SBM(n, p, W)$ with an accuracy of $\alpha \in [0, 1]$, if it outputs a labelling of the nodes $\{\sigma'(v), v \in V\}$, which agrees with the true labelling $\sigma$ on a fraction $\alpha$ of the nodes with probability $1 - o_n(1)$. The agreement is maximized over relabellings of the communities. Strong recovery refers to $\alpha = 1 - o_n(1)$ and exact recovery refers to $\alpha = 1$.*

The problem is solvable information-theoretically if there exists an algorithm that solves it, and efficiently if the algorithm runs in polynomial-time in $n$. Note that exact recovery in $SBM(n, p, W)$ requires the graph not to have vertices of degree 0 in multiple communities with high probability. Therefore, for exact recovery, we focus on $W = \ln(n)Q/n$ where $Q$ is fixed.

**I. Partial and strong recovery in the general SBM.** The first result of [1] concerns the regime where the connectivity matrix $W$ scales as $Q/n$ for a positive symmetric matrix $Q$ (i.e., the node

average degree is constant). The following notion of SNR is first introduced

$$\text{SNR} = |\lambda_{\min}|^2 / \lambda_{\max} \tag{1}$$

where $\lambda_{\min}$ and $\lambda_{\max}$ are respectively the smallest[3] and largest eigenvalues of $\text{diag}(p)Q$. The algorithm `Sphere-comparison` is proposed that solves partial recovery with exponential accuracy and quasi-linear complexity when the SNR diverges.

**Theorem 1.** *[1] Given any $k \in \mathbb{Z}$, $p \in (0,1)^k$ with $|p| = 1$, and symmetric matrix $Q$ with no two rows equal, let $\lambda$ be the largest eigenvalue of $PQ$, and $\lambda'$ be the eigenvalue of $PQ$ with the smallest nonzero magnitude. If $\text{SNR} := \frac{|\lambda'|^2}{\lambda} > 4$, $\lambda^7 < (\lambda')^8$, and $4\lambda^3 < (\lambda')^4$, for some $\varepsilon = \varepsilon(\lambda, \lambda')$ and $C = C(p,Q) > 0$, `Sphere-comparison` detects communities in graphs drawn from $SBM(n, p, Q/n)$ with accuracy $1 - 4ke^{-\frac{C\rho}{16k}}/(1 - exp(-\frac{C\rho}{16k}\left(\frac{(\lambda')^4}{\lambda^3} - 1\right)))$, provided that the above is larger than $1 - \frac{\min_i p_i}{2\ln(4k)}$, and runs in $O(n^{1+\epsilon})$ time. Moreover, $\varepsilon$ can be made arbitrarily small with $8\ln(\lambda\sqrt{2}/|\lambda'|)/\ln(\lambda)$, and $C(p, \alpha Q)$ is independent of $\alpha$.*

Note that for $k$ symmetric clusters, SNR reduces to $\frac{(a-b)^2}{k(a+(k-1)b)}$, which is the quantity of interest for detection [2, 26]. Moreover, the SNR must diverge to ensure strong recovery in the symmetric case [1]. The following is an important consequence of the previous theorem, stating that `Sphere-comparison` solves strong recovery when the entries of $Q$ are amplified.

**Corollary 1.** *[1] For any $k \in \mathbb{Z}$, $p \in (0,1)^a$ with $|p| = 1$, and symmetric matrix $Q$ with no two rows equal, there exist $\epsilon(c) = O(1/\ln(c))$ such that for all sufficiently large $c$, `Sphere-comparison` detects communities in $SBM(n, p, cQ/n)$ with accuracy $1 - e^{-\Omega(c)}$ and complexity $O_n(n^{1+\epsilon(c)})$.*

The above gives the optimal scaling both in accuracy and complexity.

**II. Exact recovery in the general SBM.** The second result in [1] is for the regime where the connectivity matrix scales as $\ln(n)Q/n$, $Q$ independent of $n$, where it is shown that exact recovery has a sharp threshold characterized by the divergence function

$$D_+(f,g) = \max_{t \in [0,1]} \sum_{x \in [k]} \left( tf(x) + (1-t)g(x) - f(x)^t g(x)^{1-t} \right),$$

named the CH-divergence in [1]. Specifically, if all pairs of columns in $\text{diag}(p)Q$ are at $D_+$-distance at least 1 from each other, then exact recovery is solvable in the general SBM. We refer to Section 2.3 in [1] for discussion on the connection with Shannon's channel coding theorem (and CH vs. KL divergence). An algorithm (`Degree-profiling`) is also developed in [1] that solves exact recovery down to the $D_+$ limit in quasi-linear time, showing that exact recovery has no informational to computational gap.

**Theorem 2.** *[1] (i) Exact recovery is solvable in $SBM(n, p, \ln(n)Q/n)$ if and only if*

$$\min_{i,j \in [k], i \neq j} D_+((PQ)_i \| (PQ)_j) \geq 1.$$

**(ii)** *The* `Degree-profiling` *algorithm (see [1]) solves exact recovery whenever it is information-theoretically solvable and runs in $o(n^{1+\epsilon})$ time for all $\epsilon > 0$.*

Exact and strong recovery are thus solved for the general SBM with linear-size communities, when the parameters are known. We next remove the latter assumption.

## 1.2 Estimating the parameters

For the estimation of the parameters, some results are known for two-symmetric communities. In the logarithmic degree regime, since the SDP is agnostic to the parameters (it is a relaxation of the min-bisection), the parameters can be estimated by recovering the communities [5, 20, 21]. For the constant-degree regime, [26] shows that the parameters can be estimated above the threshold by counting cycles (which is efficiently approximated by counting non-backtracking walks). These are, however, for 2 communities. We also became aware of a parallel work [27], which considers private graphon estimation (including SBMs). In particular, for the logarithmic degree regime, [27] obtains a (non-efficient) procedure to estimate parameters of graphons in an appropriate version of the $L_2$ norm. For the general SBM, learning the model was to date mainly open.

## 2 Results

Agnostic algorithms are developed for the constant and diverging node degrees (with $p, k$ independent of $n$). These afford optimal accuracy and complexity scaling for large node degrees and achieve the CH-divergence limit for logarithmic node degrees. In particular, the SBM can be learned efficiently for any diverging degrees.

Note that the assumptions on $p$ and $k$ being independent of $n$ could be slightly relaxed, for example to slowly growing $k$, but we leave this for future work.

### 2.1 Partial recovery

Our main result for partial recovery holds in the constant degree regime and requires a lower bound $\delta$ on the least relative size of the communities. This requirement is removed when working with diverging degrees, as stated in the corollary below.

**Theorem 3.** *Given $\delta > 0$ and for any $k \in \mathbb{Z}$, $p \in (0,1)^k$ with $\sum p_i = 1$ and $0 < \delta \leq \min p_i$, and any symmetric matrix $Q$ with no two rows equal such that every entry in $Q^k$ is positive (in other words, $Q$ such that there is a nonzero probability of a path between vertices in any two communities in a graph drawn from $SBM(n, p, Q/n)$), there exist $\epsilon(c) = O(1/\ln(c))$ such that for all sufficiently large $\alpha$,* Agnostic-sphere-comparison *detects communities in graphs drawn from $SBM(n, p, \alpha Q/n)$ with accuracy at least $1 - e^{-\Omega(\alpha)}$ in $O_n(n^{1+\epsilon(\alpha)})$ time.*

Note that a vertex in community $i$ has degree 0 with probability exponential in $c$, and there is no way to differentiate between vertices of degree 0 from different communities. So, an error rate that decreases exponentially with $c$ is optimal. In [28], we provide a more detailed version of this theorem, which yields a quantitate statement on the accuracy of the algorithm in terms of the SNR $(\lambda')^2/\lambda$ for general $SBM(n, p, Q/n)$.

**Corollary 2.** *If $\alpha = \omega(1)$ in Theorem 3, the knowledge requirement on $\delta$ can be removed.*

### 2.2 Exact recovery

Recall that from [1], exact recovery is information-theoretically and computationally solvable in $SBM(n, p, \ln(n)Q/n)$ if and only if,

$$\min_{i<j} D_+((PQ)_i, (PQ)_j) \geq 1. \tag{2}$$

We next show that this can be achieved without any knowledge on the parameters for $SBM(n, p, \ln(n)Q/n)$.

**Theorem 4.** *The* Agnostic-degree-profiling *algorithm (see Section 3.2) solves exact recovery in any $SBM(n, p, \ln(n)Q/n)$ for which exact recovery is solvable, using no input except the graph in question, and runs in $o(n^{1+\epsilon})$ time for all $\epsilon > 0$. In particular, exact recovery is efficiently and universally solvable whenever it is information-theoretically solvable.*

## 3 Proof Techniques and Algorithms

### 3.1 Partial recovery and the Agnostic-sphere-comparison algorithm

#### 3.1.1 Simplified version of the algorithm for the symmetric case

To ease the presentation of the algorithm, we focus first on the symmetric case, i.e., the SBM with $k$ communities of relative size $1/k$, probability of connecting $\frac{a}{n}$ inside communities and $\frac{b}{n}$ across communities. Let $d = (a + (k-1)b)/k$ be the average degree.

**Definition 2.** *For any vertex $v$, let $N_{r[G]}(v)$ be the set of all vertices with shortest path in $G$ to $v$ of length $r$. We often drop the subscript $G$ if the graph in question is the original SBM. We also refer to $\bar{N}_r(v)$ as the vector whose $i$-th entry is the number of vertices in $N_r(v)$ that are in community $i$.*

For an arbitrary vertex $v$ and reasonably small $r$, there will be typically about $d^r$ vertices in $N_r(v)$, and about $(\frac{a-b}{k})^r$ more of them will be in $v$'s community than in each other community. Of course,

this only holds when $r < \log n / \log d$ because there are not enough vertices in the graph otherwise. The obvious way to try to determine whether or not two vertices $v$ and $v'$ are in the same community is to guess that they are in the same community if $|N_r(v) \cap N_r(v')| > d^{2r}/n$ and different communities otherwise. Unfortunately, whether or not a vertex is in $N_r(v)$ is not independent of whether or not it is in $N_r(v')$, which compromises this plan. Instead, we propose to rely on the following **graph-splitting** step: Randomly assign every edge in $G$ to some set $E$ with a fixed probability $c$ and then count the number of edges in $E$ that connect $N_{r[G \backslash E]}$ and $N_{r'[G \backslash E]}$. Formally:

**Definition 3.** *For any $v, v' \in G$, $r, r' \in \mathbb{Z}$, and subset of $G$'s edges $E$, let $N_{r,r'[E]}(v \cdot v')$ be the number of pairs $(v_1, v_2)$ such that $v_1 \in N_{r[G \backslash E]}(v)$, $v_2 \in N_{r'[G \backslash E]}(v')$, and $(v_1, v_2) \in E$.*

Note that $E$ and $G \backslash E$ are disjoint. However, in $\mathrm{SBM}(n, p, Q/n)$, $G$ is sparse enough that even if the two graphs were generated independently, a given pair of vertices would have an edge in both graphs with probability $O(\frac{1}{n^2})$. So, $E$ is approximately independent of $G \backslash E$.

Thus, given $v, r$, and denoting by $\lambda_1 = (a + (k-1)b)/k$ and $\lambda_2 = (a-b)/k$ the two eigenvalues of $PQ$ in the symmetric case, the expected number of intra-community neighbors at depth $r$ from $v$ is approximately $\frac{1}{k}(\lambda_1^r + (k-1)\lambda_2^r)$, whereas the expected number of extra-community neighbors at depth $r$ from $v$ is approximately $\frac{1}{k}(\lambda_1^r - \lambda_2^r)$ for each of the other $(k-1)$ communities. All of these are scaled by $1 - c$ if we do the computations in $G \backslash E$. Using now the emulated independence between $E$ and $G \backslash E$, and assuming $v$ and $v'$ to be in the same community, the expected number of edges in $E$ connecting $N_{r[G \backslash E]}(v)$ to $N_{r'[G \backslash E]}(v')$ is approximately given by the inner product $u^t(c \cdot PQ)u$, where $u = \frac{1}{k}(\lambda_1^r + (k-1)\lambda_2^r, \lambda_1^r - \lambda_2^r, \ldots, \lambda_1^r - \lambda_2^r)$ and $(PQ)$ is the matrix with $a$ on the diagonal and $b$ elsewhere. When $v$ and $v'$ are in different communities, the inner product is between $u$ and a permutation of $u$. After simplifications, this gives

$$N_{r,r'[E]}(v \cdot v') \approx \frac{c(1-c)^{r+r'}}{n} \left[ d^{r+r'+1} + \left(\frac{a-b}{k}\right)^{r+r'+1} (k\delta_{\sigma_v, \sigma_{v'}} - 1) \right] \quad (3)$$

where $\delta_{\sigma_v, \sigma_{v'}}$ is 1 if $v$ and $v'$ are in the same community and 0 otherwise. In order for $N_{r,r'[E]}(v \cdot v')$ to depend on the relative communities of $v$ and $v'$, it must be that $c(1-c)^{r+r'} |\frac{a-b}{k}|^{r+r'+1} k$ is large enough, i.e., more than $n$, so $r + r'$ needs to be at least $\log n / \log |\frac{a-b}{k}|$. A difficulty is that for a specific pair of vertices, the $d^{r+r'+1}$ term will be multiplied by a random factor dependent on the degrees of $v, v'$, and the nearby vertices. So, in order to stop the variation in the $d^{r+r'+1}$ term from drowning out the $\left(\frac{a-b}{k}\right)^{r+r'+1} (k\delta_{\sigma_v, \sigma_{v'}} - 1)$ term, it is necessary to cancel out the dominant term. This brings us to introduce the following **sign-invariant statistics**:

$$I_{r,r'[E]}(v \cdot v') := N_{r+2,r'[E]}(v \cdot v') \cdot N_{r,r'[E]}(v \cdot v') - N_{r+1,r'[E]}^2(v \cdot v')$$

$$\approx \frac{c^2(1-c)^{2r+2r'+2}}{n^2} \cdot \left(d - \frac{a-b}{k}\right)^2 \cdot d^{r+r'+1} \left(\frac{a-b}{k}\right)^{r+r'+1} (k\delta_{\sigma_v, \sigma_{v'}} - 1)$$

In particular, for $r + r'$ odd, $I_{r,r'[E]}(v \cdot v')$ will tend to be positive if $v$ and $v'$ are in the same community and negative otherwise, irrespective of the specific values of $a, b, k$. That suggests the following algorithm for partial recovery, it requires knowledge of $\delta < 1/k$ in the constant degree regime, but not in the regime where $a, b$ scale with $n$.

---

1. Set $r = \frac{3}{4} \log n / \log d$ and put each of the graph's edges in $E$ with probability $1/10$.
2. Set $k_{\max} = 1/\delta$ and select $k_{\max} \ln(4k_{\max})$ random vertices, $v_1, ..., v_{k_{\max} \ln(4k_{\max})}$.
3. Compute $I_{r,r'[E]}(v_i \cdot v_j)$ for each $i$ and $j$.
4. If there is a possible assignment of these vertices to communities such that $I_{r,r'[E]}(v_i \cdot v_j) > 0$ if and only if $v_i$ and $v_j$ are in the same community, then randomly select one vertex from each apparent community, $v[1], v[2], ...v[k']$. Otherwise, fail.
5. For every $v'$ in the graph, guess that $v'$ is in the same community as the $v[i]$ that maximizes the value of $I_{r,r'[E]}(v[i] \cdot v')$.

---

This algorithm succeeds as long as $|a - b|/k > (10/9)^{1/6}((a + (k-1)b)/k)^{5/6}$, to ensure that the above estimates on $N_{r,r'[E]}(v \cdot v')$ are reliable. Further, if $a, b$ are scaled by $\alpha = \omega(1)$, setting $\delta = 1/\log\log\alpha$ allows removal of the knowledge requirement on $\delta$.

One alternative to our approach could be to count the non-backtracking walks of a given length between $v$ and $v'$, like in [4, 29], instead of using $N_{r,r'[E]}(v \cdot v')$. However, proving that the number of non-backtracking walks is close to its expected value is difficult. Proving that $N_{r,r'[E]}(v \cdot v')$ is within a desired range is substantially easier because for any $v_1$ and $v_2$, whether or not there is an edge between $v_1$ and $v_2$ directly effects $N_r(v)$ for at most one value of $r$. Algorithms based on shortest path have also been studied in [30].

### 3.1.2 The general case

In the general case, define $N_r(v)$, $\bar{N}_r(v)$ and $N_{r,r'[E]}(v \cdot v')$ as in the previous section. Now, for any $v_1 \in N_{r[G/E]}(v)$ and $v_2 \in N_{r'[G/E]}(v')$, $(v_1, v_2) \in E$ with a probability of approximately $cQ_{\sigma_{v_1}, \sigma_{v_2}}/n$. As a result,

$$N_{r,r'}[E](v \cdot v') \approx \bar{N}_{r[G\backslash E]}(v) \cdot \frac{cQ}{n}\bar{N}_{r'[G\backslash E]}(v') \approx ((1-c)PQ)^r e_{\sigma_v} \cdot \frac{cQ}{n}((1-c)PQ)^{r'} e_{\sigma_{v'}}$$

$$= c(1-c)^{r+r'} e_{\sigma_v} \cdot Q(PQ)^{r+r'} e_{\sigma_{v'}}/n.$$

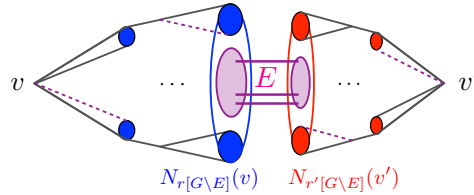

$$N_{r[G\backslash E]}(v) \qquad N_{r'[G\backslash E]}(v')$$

Figure 1: The purple edges represent the edges counted by $N_{r,r'}[E](v \cdot v')$.

Let $\lambda_1, ..., \lambda_h$ be the distinct eigenvalues of $PQ$, ordered so that $|\lambda_1| \geq |\lambda_2| \geq ... \geq |\lambda_h| \geq 0$. Also define $h'$ so that $h' = h$ if $\lambda_h \neq 0$ and $h' = h - 1$ if $\lambda_h = 0$. If $W_i$ is the eigenspace of $PQ$ corresponding to the eigenvalue $\lambda_i$, and $P_{W_i}$ is the projection operator on to $W_i$, then

$$N_{r,r'}[E](v \cdot v') \approx c(1-c)^{r+r'} e_{\sigma_v} \cdot Q(PQ)^{r+r'} e_{\sigma_{v'}}/n \qquad (4)$$

$$= \frac{c(1-c)^{r+r'}}{n} \sum_i \lambda_i^{r+r'+1} P_{W_i}(e_{\sigma_v}) \cdot P^{-1} P_{W_i}(e_{\sigma_{v'}}) \qquad (5)$$

where the final equality holds because for all $i \neq j$,

$$\lambda_i P_{W_i}(e_{\sigma_v}) \cdot P^{-1} P_{W_j}(e_{\sigma_{v'}}) = (PQP_{W_i}(e_{\sigma_v})) \cdot P^{-1} P_{W_j}(e_{\sigma_{v'}})$$

$$= P_{W_i}(e_{\sigma_v}) \cdot QP_{W_j}(e_{\sigma_{v'}}) = P_{W_i}(e_{\sigma_v}) \cdot P^{-1} \lambda_j P_{W_j}(e_{\sigma_{v'}}),$$

and since $\lambda_i \neq \lambda_j$, this implies that $P_{W_i}(e_{\sigma_v}) \cdot P^{-1} P_{W_j}(e_{\sigma_{v'}}) = 0$.

**Definition 4.** Let $\zeta_i(v \cdot v') = P_{W_i}(e_{\sigma_v}) \cdot P^{-1} P_{W_i}(e_{\sigma_{v'}})$ for all $i$, $v$, and $v'$.

Equation (5) is dominated by the $\lambda_1^{r+r'+1}$ term, so getting good estimate of the $\lambda_2^{r+r'+1}$ through $\lambda_{h'}^{r+r'+1}$ terms requires cancelling it out somehow. As a start, if $\lambda_1 > \lambda_2 > \lambda_3$ then

$$N_{r+2,r'[E]}(v \cdot v') \cdot N_{r,r'[E]}(v \cdot v') - N_{r+1,r'[E]}^2(v \cdot v')$$

$$\approx \frac{c^2(1-c)^{2r+2r'+2}}{n^2}(\lambda_1^2 + \lambda_2^2 - 2\lambda_1\lambda_2)\lambda_1^{r+r'+1}\lambda_2^{r+r'+1}\zeta_1(v \cdot v')\zeta_2(v \cdot v')$$

Note that the left hand side of this expression is equal to $\det \begin{vmatrix} N_{r,r'[E]}(v \cdot v') & N_{r+1,r'[E]}(v \cdot v') \\ N_{r+1,r'[E]}(v \cdot v') & N_{r+2,r'[E]}(v \cdot v') \end{vmatrix}$.

**Definition 5.** Let $M_{m,r,r'[E]}(v \cdot v')$ be the $m \times m$ matrix such that $M_{m,r,r'[E]}(v \cdot v')_{i,j} = N_{r+i+j,r'[E]}(v \cdot v')$ for each $i$ and $j$.

As shown in [28], there exists constant $\gamma(\lambda_1, ..., \lambda_m)$ such that

$$\det(M_{m,r,r'[E]}(v \cdot v')) \approx \frac{c^m(1-c)^{m(r+r')}}{n^m}\gamma(\lambda_1, ..., \lambda_m)\prod_{i=1}^{m}\lambda_i^{r+r'+1}\zeta_i(v \cdot v') \qquad (6)$$

where we assumed that $|\lambda_m| > |\lambda_{m+1}|$ above to simplify the discussion (the case $|\lambda_m| = |\lambda_{m+1}|$ is similar). This suggests the following plan for estimating the eigenvalues corresponding to a graph. First, pick several vertices at random. Then, use the fact that $|N_{r[G\setminus E]}(v)| \approx ((1-c)\lambda_1)^r$ for any good vertex $v$ to estimate $\lambda_1$. Next, take ratios of (6) for $m$ and $m-1$ (with $r = r'$), and look for the smallest $m$ making that ratio small enough (this will use the estimate on $\lambda_1$), estimating $h'$ by this value minus one. Then estimate consecutively all of $PQ$'s eigenvalues for each selected vertex using ratios of (6). Finally, take the median of these estimates.

In general, whether $|\lambda_m| > |\lambda_{m+1}|$ or $|\lambda_m| = |\lambda_{m+1}|$,

$$\frac{\det(M_{m,r+1,r'[E]}(v \cdot v')) - (1-c)^m\lambda_{m+1}\prod_{i=1}^{m-1}\lambda_i\det(M_{m,r,r'[E]}(v \cdot v'))}{\det(M_{m-1,r+1,r'[E]}(v \cdot v')) - (1-c)^{m-1}\lambda_m\prod_{i=1}^{m-2}\lambda_i\det(M_{m-1,r,r'[E]}(v \cdot v'))}$$

$$\approx \frac{c}{n}\frac{\gamma(\lambda_1, ..., \lambda_m)}{\gamma(\lambda_1, ..., \lambda_{m-1})}\frac{\lambda_{m-1}(\lambda_m - \lambda_{m+1})}{\lambda_m(\lambda_{m-1} - \lambda_m)}((1-c)\lambda_m)^{r+r'+2}\zeta_m(v \cdot v').$$

This fact can be used to approximate $\zeta_i(v \cdot v')$ for arbitrary $v$, $v'$, and $i$. Of course, this requires $r$ and $r'$ to be large enough that $\frac{c(1-c)^{r+r'}}{n}\lambda_i^{r+r'+1}\zeta_i(v \cdot v')$ is large relative to the error terms for all $i \le h'$. This requires at least $|(1-c)\lambda_i|^{r+r'+1} = \omega(n)$ for all $i \le h'$. Moreover, for any $v$ and $v'$,

$$0 \le P_{W_i}(e_{\sigma_v} - e_{\sigma_{v'}}) \cdot P^{-1}P_{W_i}(e_{\sigma_v} - e_{\sigma_{v'}}) = \zeta_i(v \cdot v) - 2\zeta_i(v \cdot v') + \zeta_i(v' \cdot v')$$

with equality for all $i$ if and only if $\sigma_v = \sigma_{v'}$, so sufficiently good approximations of $\zeta_i(v \cdot v), \zeta_i(v \cdot v')$ and $\zeta_i(v' \cdot v')$ can be used to determine which pairs of vertices are in the same community.

One could generate a reasonable classification based solely on this method of comparing vertices (with an appropriate choice of the parameters, as later detailed). However, that would require computing $N_{r,r'[E]}(v \cdot v)$ for every vertex in the graph with fairly large $r + r'$, which would be slow. Instead, we use the fact that for any vertices $v$, $v'$, and $v''$ with $\sigma_v = \sigma_{v'} \ne \sigma_{v''}$,

$$\zeta_i(v' \cdot v') - 2\zeta_i(v \cdot v') + \zeta_i(v \cdot v) = 0 \le \zeta_i(v'' \cdot v'') - 2\zeta_i(v \cdot v'') + \zeta_i(v \cdot v)$$

for all $i$, and the inequality is strict for at least one $i$. So, subtracting $\zeta_i(v \cdot v)$ from both sides,

$$\zeta_i(v' \cdot v') - 2\zeta_i(v \cdot v') \le \zeta_i(v'' \cdot v'') - 2\zeta_i(v \cdot v'')$$

for all $i$, and the inequality is still strict for at least one $i$. So, given a representative vertex in each community, we can determine which of them a given vertex, $v$, is in the same community as without needing to know the value of $\zeta_i(v \cdot v)$.

This runs fairly quickly if $r$ is large and $r'$ is small because the algorithm only requires focusing on $|N_{r'}(v')|$ vertices. This leads to the following plan for partial recovery. First, randomly select a set of vertices that is large enough to contain at least one vertex from each community with high probability. Next, compare all of the selected vertices in an attempt to determine which of them are in the same communities. Then, pick one in each community. Call these **anchor** nodes. After that, use the algorithm referred to above to determine which community each of the remaining vertices is in. As long as there actually was at least one vertex from each community in the initial set and none of the approximations were particularly bad, this should give a reasonable classification. The risk that this randomly gives a bad classification due to a bad set of initial vertices can be mitigated by repeating the previous classification procedure several times as discussed in [28]. This completes the `Agnostic-sphere-comparison` algorithm. We refer to [28] for the details.

## 3.2   Exact recovery and the `Agnostic-degree-profiling` **algorithm**

The exact recovery part is similar to [1] and uses the fact that once a good enough clustering has been obtained from `Agnostic-sphere-comparison`, the classification can be finished by making local improvements based on the node's neighborhoods. Similar techniques have been used in [5, 11, 19, 31, 32]. However, we establish here a sharp characterization of the local procedure error.

The key result is that, when testing between two multivariate Poisson distributions of means $\log(n)\lambda_1$ and $\log(n)\lambda_2$ respectively, where $\lambda_1, \lambda_2 \in \mathbb{Z}_+^k$, the probability of error (of maximum a posteriori decoding) is

$$n^{-D_+(\lambda_1, \lambda_2) + o(1)}. \tag{7}$$

This is proved in [1]. In the case of unknown parameters, the algorithmic approach is largely unchanged, adding a step where the best known classification is used to estimate $p$ and $Q$ prior to any local improvement step. The analysis of the algorithm requires however some careful handling.

First, it is necessary to prove that given a labelling of the graph's vertices with an error rate of $x$, one can compute approximations of $p$ and $Q$ that are within $O(x + \log(n)/\sqrt{n})$ of their true values with probability $1 - o(1)$. Secondly, one needs to modify the above hypothesis testing estimates to control the error probability. In attempting to determine vertices' communities based on estimates of $p$ and $Q$ that are off by at most $\delta$, say $p'$ and $Q'$, one must show that a classification of its neighbors that has an error rate of $\delta$ classifies the vertices with an error rate only $e^{O(\delta \log n)}$ times higher than it would be if the parameter really were $p'$ and $Q'$ and the vertices' neighbors were all classified correctly. Thirdly, one needs to show that since $D_+((PQ)_i, (PQ)_j)$ is differentiable with respect to any element of $PQ$, the error rate if the parameters really were $p'$ and $Q'$ is at worst $e^{O(\delta \log n)}$ as high as the error rate with the actual parameters. Combining these yields the conclusion that any errors in the estimates of the SBM's parameters do not disrupt vertex classification any worse than the errors in the preliminary classifications already were.

**The** `Agnostic-degree-profiling` **algorithm.** The inputs are $(G, \gamma)$, where $G$ is a graph, and $\gamma \in [0, 1]$ (see [28] for how to set $\gamma$ specifically). The algorithm outputs each node's label.

(1) Define the graph $g'$ on the vertex set $[n]$ by selecting each edge in $g$ independently with probability $\gamma$, and define the graph $g''$ that contains the edges in $g$ that are not in $g'$.
(2) Run `Agnostic-sphere-comparison` on $g'$ with $\delta = 1/\log\log(n)$ to obtain the classification $\sigma' \in [k]^n$.
(3) Determine the size of each alleged community, and the edge density between each pair of alleged communities.
(4) For each node $v \in [n]$, determine the most likely community label of node $v$ based on its degree profile $\bar{N}_1(v)$ computed from the preliminary classification $\sigma'$, and call it $\sigma_v''$.
(5) Use $\sigma_v''$ to get new estimates of $p$ and $Q$.
(6) For each node $v \in [n]$, determine the most likely community label of node $v$ based on its degree profile $\bar{N}_1(v)$ computed from $\sigma''$. Output this labelling.

In step (3) and (6), the most likely label is the one that maximizes the probability that the degree profile comes from a multivariate distribution of mean $\ln(n)(PQ)_i$ for $i \in [k]$. Note that this algorithm does not require a lower bound on $\min p_i$ because setting $\delta$ to a slowly decreasing function of $n$ results in $\delta$ being within an acceptable range for all sufficiently large $n$.

## 4  Data implementation and open problems

We tested a simplified version of our algorithm on real data (see [28]), for the blog network of Adamic and Glance '05. We obtained an error rate of about 60/1222 (best trial was 57, worst 67), achieving the state-of-the-art (as described in [32]). The results in this paper should extend quite directly to a slowly growing number of communities (e.g., up to logarithmic). It would be interesting to extend the current approach to smaller sized or more communities, watching the complexity scaling, as well as to corrected-degrees, labeled-edges, or overlapping communities (though the approach in this paper already applies to linear-sized overlaps).

## Footnotes

[1] [6] generalizes this to $a, b = \Theta(1)$.

[2] [24] also study variations of the $k$-symmetric model.

[3]The smallest eigenvalue of $\text{diag}(p)Q$ is the one with least magnitude.

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
