[Reviews · NeurIPS 2015]

Submitted by Assigned_Reviewer_1

This paper seems to provide new and deep results on recovering the cluster structure in the stochastic block model even if the number of clusters is unknown. However, once again, a possibly very good paper is impaired by quite bad writing (hence the 7 on QS). For instance, the notations needed to understand the results recalled in Section 1.1 are introduced in Section 2.1. Why would anyone do that? In addition the paper comes with 47 pages of supplementary material which shows that the NIPS submission is a summary of a full journal paper. I'm not sure NIPS is the correct venue for such submission, especially considering that it's in my opinion impossible to reconstruct the algorithms proposed in the paper based on the vague description it contains. One has to rely on the supplementary material to understand the algorithm.
Summary: Probably a very good paper, but badly written and not very adapted to the NIPS format (47 pages of supplementary material!).

Submitted by Assigned_Reviewer_2

1) The paper is not readable for the format of this conference. It can be submitted to a journal.

2) The authors disclose their names in the supplementary material at the beginning of the page 13.
Summary: This paper has similarities to the paper: "Community detection in general stochastic block models: fundamental limits and efficient algorithms". The authors also disclose their names in the supplementary material, claiming the authorship of this paper.

Submitted by Assigned_Reviewer_3

[light review] The authors might be interested in the paper "Achieving optimal misclassification proportion in the stochastic blockmodel" http://arxiv.org/abs/1505.03772, and also "minimax rates of community detection in stochastic blockmodels" http://arxiv.org/abs/1507.05313
Summary: [light review] Gives efficient algorithm to achieve optimal classification rate in stochastic blockmodel, this is currently an active area of research with many deep results. However, the paper is not very readable. There isn't even a clear description of the algorithms, although there might be more in the supplemental material. It feels like only minimal effort was spent converting a long paper into a shorter one.

Submitted by Assigned_Reviewer_4

Typically, SBMs require a number of communities to be specified beforehand.

The proposed algorithms for SBM does not require the information, but achieve an optimal information-theoretic tradeoffs.

The authors prove accuracies and conditions for partial and exact recoveries.

I need to be honest that I cannot fully follow the meaning of theorems and therefore cannot fully understand the contributions of the paper.

I'm working on community clustering in a private company thus my viewpoint is weighed on a practical side for engineers.

Theorem 3 and 4: What are the improvements from the known results? It is very helpful if you can provide intuitive explanations for readers unfamiliar for the SBM theories (like me).

Line 228: You need to know"the relative sizes of communities", don't you?

I think the information about "the relative sizes of communities" is difficult to determine (know) beforehand like the number of clusters. Assume a social network that typically has a long-tailed community structure. It is difficult to determine the finest or the largest size of a community.

Line 419-420: Only one real-world data test is no enough for a proof of concept. I am rather interested in how robust the proposed algorithm is for various real networks that does not necessarily satisfy assumptions.

How about the computational cost? Works in realistic computational time for large networks?
Summary: Please find the detailed comment.

Submitted by Assigned_Reviewer_5

The paper presents two algorithms for community recovery of nodes for graphs generated from stochastic block models (SBM). Two different regimes of SBM has been considered, one with constant average degree and and another with average degree of the order log(n). For the two cases, algorithms have been proposed based on shortest path finding. Main results of the paper are

bounds on partial recovery of node labels with high probability for sparse (constant degree) regime and complete recovery of node labels in semi-sparse (log(n)-average degree) regime above the information-theoretic threshold of partial and complete recovery. The error bounds are given based on the proposed algorithms and shown to be optimal. The paper builds on a previous paper (Abbe and Sanders, 2015), which presented similar algorithms and analysis but with the restriction that knowledge of model parameters were needed. In this paper those restrictions are not needed anymore.

The paper along with the supplementary materials is quite mathematically rigorous. Technically, the paper is of high quality. It presents strong results on almost exact recovery and exact recovery of community labels without assuming knowledge of model parameters. However, the paper lacks a bit in numerical comparison of performance with other methods of community detection.

However, the main drawback of the paper is clarity. The paper presents two different algorithms, but, the algorithms are not properly outlined in the paper. The bulk of the paper including the main algorithm statements is in the supplementary material and without reading through the supplementary material, it is very difficult to parse the paper. The paper reads like a journal paper which has been truncated to meet the page constraints. The agnostic-sphere-comparison algorithm, which is the main algorithm of the paper, need to be presented a bit more clearly, like the way it has been presented in the supplementary material. Also, note that definitions of SBM is not consistent between Theorem 1 and Corollary 1 (one use Q and another Q/n) and and the way SBM has been defined states Q as probability (in line 47, page 1), which is inconsistent with the normalizations of Q made later in the paper.

The paper presents an original algorithm of approximately linear time and gives a mathematically rigorous proof that the algorithm works for community recovery for general SBM. This makes the algorithms presented in the paper as one of the state-of-the-art methods for community recovery. Algorithms based on shortest paths has been previously proposed in literature (Bhattacharyya and Bickel, 2014), but in the current paper the algorithm is much more subtle and the results obtained are tighter.

The paper is significant in terms of the information theoretic bounds presented in the paper as well as the algorithms introduced. The paper can be generalized to analyze more general network models, specially for growing number of communities. But, due to the technical subtlety of the paper, I feel that the paper is more suited for a journal publication than a short conference publication.
Summary: The paper is a summarized version of a technical paper with some interesting results on recovery of community identity of nodes for graphs generated from stochastic block models. The paper is technically strong but lacks in clarity in the presentation of main methods of the paper.

Author Feedback
Author rebuttal: In response to reviewer 1: we are sorry about the involuntary leakage in the supplementary material, but isn't this leakage superficial in the case of this paper, as it is obvious throughout the entire paper that the authors are the same as in [AS15]? On the other hand, we strongly disagree with the two other statements made by reviewer 1. The NIPS submission goes far beyond [AS15], and at no time the paper tries to hide or minimize the overlap between our NIPS submission and [AS15]:

[AS15] (which will appear in FOCS '15) requires full knowledge of the parameters of the SBM, that is, knowledge of the number of communities, their relative sizes and the connectivity parameters. The results of [AS15] establish the first *necessary and sufficient* condition for recovering communities in the general SBM (and the condition is explicitly given in terms of a new f-divergence). However, the requirement of knowing the parameters is a serious weakness of [AS15], which makes [AS15] possibly more of a mathematical than practical result. In our NIPS submission, we completely revised our approach and proposed a new algorithm that does not require any knowledge. The new approach is based on comparing invariants of the statistics of common neighbors in the network. These invariants are given by the determinant of Vandermonde matrices formed by eigenvalues of the SBM connectivity matrix. This is a completely new approach which is likely to start a new research direction for community detection.

Is removing the parameter knowledge an important extension? This is where we are very surprised about reviewer 1's comment. It is well admitted that one of the hardest problems in community detection is to learn the number of communities. Prior to our NIPS submission, there were no rigorous results to learn the number of communities, and we have now solved this problem for the SBM. An alternative seems to be the recent results on the non-backtracking operator that allows (perhaps) one to guess the number of communities when raising the offset of the perturbed adjacency matrix; however this approach has not yet been shown to succeed. Nor have other spectral methods, or SDPs. To the best of our knowledge, our NIPS submission is the first paper that allows to learn the number of communities and all other parameters in one shot and with guarantees. For these reasons, we believe that our submission is far from a small (or moderate) extension of [AS15].

In addition, hard work was needed to make our algorithm run in quasi-linear time; this is far from obvious as any easy-to-analyze method is at least quadratic, whereas quasi-linearity is required for large networks. This answers reviewer 2 question about complexity scaling. Regarding numerical tests, we wanted to give at least one vignette that our algorithm is practical, showing that it achieves the state of the art on the blog network. Our goal was not to compare our algorithms on all possible data sets, as this work is clearly about obtaining obtaining guarantees. We agree with reviewer 2 that more tests would be nice, but there was only so much we could do in one technical paper. We are happy to do more in the future. Note also that we do not need the smallest relative size for our main result (as reviewer 2 asks).

We thought that learning the SBM would be of interest to NIPS, and have hence decided to submit the results to NIPS. We probably did not write the paper in the best format, and apologize for that. This was our first NIPS submission, and we would have appreciated a slightly more constructive feedback from the reviewers than simply mentioning it is not well adapted to NIPS. It is absolutely wrong that we spent the minimal amount of time for the abstract. The real point is that we found it truly challenging to summarize our algorithm in few pages, and this may not be due to writing style only. A result of this generality may not be explained in a few pages easily. We do not value technical results nor long papers. However, as evidenced by the fact that spectral or SDPs or any other methods currently fall short in solving the problem that we solve, one should also recognize that there may be an intrinsic technical aspect to deal with.

We have however found a fix that could be used to improve the presentation. We could explain the algorithm for a simpler case of SBM, with equal-size communities and only 2 connectivity probabilities (inside and across communities). In such a setting, several of the technicalities appearing in the general setting go away, and we could explain the algorithm more succinctly and clearly. Perhaps we can even give up the quasi-linear complexity to further simplify the algorithm. Then we could mention the result in greater generality only at the end.

Thank you to all the reviewers for the feedback. We are glad that 4 out of 6 recommended acceptance, and trust that we can improve the presentation if a final submission is requested.